# The Influence of Electron Beam Sterilization on In Vivo Degradation of β-TCP/PCL of Different Composite Ratios for Bone Tissue Engineering

**DOI:** 10.3390/mi11030273

**Published:** 2020-03-06

**Authors:** Jin-Ho Kang, Janelle Kaneda, Jae-Gon Jang, Kumaresan Sakthiabirami, Elaine Lui, Carolyn Kim, Aijun Wang, Sang-Won Park, Yunzhi Peter Yang

**Affiliations:** 1Department of Prosthodontics, School of Dentistry, Chonnam National University, Gwanju 61186, Korea; jhk.bme1002@gmail.com (J.-H.K.); jangjaegon@naver.com (J.-G.J.); sakthikarthi.dentist@gmail.com (K.S.); 2Department of Bioengineering, Stanford University, Stanford, CA 94305, USA; jkaneda@stanford.edu; 3Department of Mechanical Engineering, Stanford University, Stanford, CA 94305, USA; elainelui@stanford.edu (E.L.); ck2514@stanford.edu (C.K.); 4Surgical Bioengineering Laboratory, Department of Surgery, School of Medicine, University of California–Davis, Sacramento, CA 95817, USA; aawang@ucdavis.edu; 5Department of Biomedical Engineering, University of California–Davis, Davis, CA 95616, USA; 6Institute for Pediatric Regenerative Medicine, Shriners Hospitals for Children–Northern California, Sacramento, CA 95817, USA; 7Department of Orthopaedic Surgery, Stanford University, Stanford, CA 94305, USA; 8Department of Materials Science and Engineering, Stanford University, Stanford, CA 94305, USA

**Keywords:** 3D printing, β-tricalcium phosphate/polycaprolactone (β-TCP/PCL) composite, bone tissue engineering, electron beam sterilization

## Abstract

We evaluated the effect of electron beam (E-beam) sterilization (25 kGy, ISO 11137) on the degradation of β-tricalcium phosphate/polycaprolactone (β-TCP/PCL) composite filaments of various ratios (0:100, 20:80, 40:60, and 60:40 TCP:PCL by mass) in a rat subcutaneous model for 24 weeks. Volumes of the samples before implantation and after explantation were measured using micro-computed tomography (micro-CT). The filament volume changes before sacrifice were also measured using a live micro-CT. In our micro-CT analyses, there was no significant difference in volume change between the E-beam treated groups and non-E-beam treated groups of the same β-TCP to PCL ratios, except for the 0% β-TCP group. However, the average volume reduction differences between the E-beam and non-E-beam groups in the same-ratio samples were 0.76% (0% TCP), 3.30% (20% TCP), 4.65% (40% TCP), and 3.67% (60% TCP). The E-beam samples generally had more volume reduction in all experimental groups. Therefore, E-beam treatment may accelerate degradation. In our live micro-CT analyses, most volume reduction arose in the first four weeks after implantation and slowed between 4 and 20 weeks in all groups. E-beam groups showed greater volume reduction at every time point, which is consistent with the results by micro-CT analysis. Histology results suggest the biocompatibility of TCP/PCL composite filaments.

## 1. Introduction

There have been many developments in engineering biocompatible and biodegradable bone implant scaffolds for use as an alternative to autografts and allografts for bone defects over the last couple of decades, especially when bone defects are large and donor morbidity is a risk [1,2,3,4]. Bone scaffolds have been extensively studied to promote native bone tissue growth and surrounding cell proliferation by optimizing nutrient transportation and mimicking native mechanical properties while minimizing damage to the surrounding tissues [1,2,5,6,7]. Bone scaffolds have been constructed using various materials such as metals, bioglasses, ceramics, and polymers, and are typically fabricated from a composite of the latter two [1,2,5,7]. Specifically, β-tricalcium phosphate (β-TCP), a polymorph of tricalcium phosphate and a biomimetic ceramic, and polycaprolactone (PCL), a biocompatible polymer, are two commonly used, clinically available biodegradable materials in bone scaffold engineering. TCP has a comparable resorption rate to bone regeneration [1,8,9]. Additionally, when compared to five other commonly used FDA-approved poly(α-hydroxy esters), PCL was one of two that demonstrated the best structural integrity and cellular response [10].

The composite construct of β-TCP and PCL combines the respective benefits of each: osteoconductivity—or bone growth on a surface such as an implant scaffold [11,12,13,14]—and easy handling, both of which have only begun to be explored in further depth. The β-TCP/PCL composite is composed of the osteoconductive β-TCP ceramic particles suspended in the bioresorbable PCL polymer matrix [15,16]. The composite material’s ability to be extruded into a filament and then 3D-printed enables the creation of controlled, patient-specific scaffolds to optimize its integration within and support for native bone tissue regeneration [17]. In addition to 3D printing, other scaffold fabrication methods include electrospinning, solvent casting, particle leaching, thermally-induced phase separation, and various molding techniques [5,6,18].

Although various factors for optimizing bone scaffolds have been studied, examining the degradation profiles of these constructs is particularly crucial for evaluating the success of bone implants for clinical applications. A bone scaffold should subsist long enough to induce the maximum therapeutic effect at the bone defect site, but also degrade when healing is underway. Poly (α-hydroxy esters)—and by association, composites with polymers in this group—undergo hydrolytic degradation via two methods: surface or bulk [19]. Ideally, degradation and resorption times for bone scaffolds should match bone regeneration rates, depending on the bone defect size. For large bone defects, the degradation and resorption duration for bone scaffolds can be greater than two years [20]. Slow-degrading scaffolds have been shown to prevent tears, allow a slow reintegration of movement, and minimize toxicity at the site of interest when compared to fast-degrading scaffolds [21].

Since sterilization is necessary for the clinical realization of a bone scaffold, it is important to then study how sterilization may change degradation, which further affects the structural integrity and mechanical profiles of bone scaffolds. Various sterilization methods exist for bone scaffolds, including heat-based ethylene oxide immersion and irradiation via ultraviolet, gamma, and electron beam (E-beam) irradiation [22]. Submersion in solvents, such as 70% ethanol, has also been used to sterilize scaffolds, but is insufficient as a sterilization method alone because ethanol has minimal sterilizing power over bacterial spores [22]. Out of all of these methods, E-beam is the most optimal for pre-packaged biomaterials with low melting points, which is relevant for β-TCP/PCL scaffolds [22]. Additionally, E-beam has higher dosage rates than both ultraviolet and gamma irradiation methods, resulting in less exposure time [22]. This is particularly important for polymers like PCL, because irradiation methods like E-beam and gamma have been shown to increase the polydispersity of PCL chains and affect mechanical properties and degradation rates [22,23,24]. This is a result of PCL ester–ester chain scissioning, in addition to crosslinking, or the formation of chemical bonds to connect polymer chains [23,24].

In our previous study, we found a 14% increase in the initial Young’s modulus and a 25% faster in vitro degradation profile for scaffolds that received E-beam compared to those that did not [23]. The increased Young’s modulus values after E-beam were likely due to crosslinking, which strengthens the β-TCP/PCL composite structure, while the increase in degradation rate after E-beam in vitro was likely due to chain scissioning, which is thought to weaken the composite structure [23]. Furthermore, since β-TCP particles are merely suspended in the polymer matrix, degradation of β-TCP/PCL scaffolds in any given solution is mainly driven by polymer degradation via the hydrolytic cleavage or scissioning of ester–ester linkages [19,20,23,25]. Previous studies, including ours, have focused solely on 20% TCP/80% PCL [19,20,23,26,27], so this study extends the work by examining the in vivo degradation profiles of various β-TCP/PCL composite ratios by mass (0:100, 20:80, 40:60, and 60:40) in a rat model, particularly studying the effect of E-beam sterilization among these different ratios on in vivo degradation. We have chosen to use extruded filament samples over scaffold samples for this in vivo study for simplification and as a screening test for chemical compositions. While we recognize that scaffolds confer additional properties, such as porosity, that can also influence degradation, the main purpose of our in vivo study is to test how the chemical composition and E-beam affect degradation. This can be achieved using extruded filament samples, while also saving time and cost. In addition, these extruded filaments can help predict the degradation of extrusion-based printed devices and grafts.

## 2. Materials and Methods

### 2.1. Sample Fabrication

The sample fabrication protocol was adapted from Bruyas et al. [28]. Four ratios of β-TCP to PCL were synthesized from the stock constituents using a protocol involving dissolution and precipitation phases. The gram-to-gram ratio of β-TCP powder with an average particle size of 100 nm (Berkeley Advanced Biomaterials Inc., Berkeley, CA, USA) to PCL pellets (Sigma-Aldrich, St. Louis, MO, USA) was 0:37.5, 7.5:30, 15:22.5, and 22.5:15 for β-TCP to PCL ratios of 0:100, 20:80, 40:60, and 60:40 by mass, respectively. Materials were suspended in dimethylformamide (DMF) (Fisher Chemical, Waltham, MA, USA): 20 mL DMF per 1 g β-TCP, and 10 mL DMF per 1 g PCL. The materials were gradually mixed into DMF separately by heating each beaker to 70–90 °C and stirring for three hours, before the two were combined and stirred for an additional hour. The mixture was then precipitated into a large container of cold tap water, flattened into a sheet of approximately 200–350 cm^2^ area, and dried at room temperature overnight. The composite material was then hand-processed into pellets with diameters of approximately 5 mm. These pellets were fed into a lab-built screw extruder to create a filament with an average diameter of approximately 2.5 mm. Ratios with higher β-TCP content required higher temperatures for extrusion, since β-TCP has a much higher melting point than PCL (1670 °C versus 60 °C, respectively). A 90 °C temperature was used for 0:100 and 20:80, while 100 °C was used for 40:60 and 120 °C for 60:40. This material- and filament-synthesis process was repeated for each of the four ratios. Samples 5 mm long were cut from each filament material for the in vivo study.

#### 2.1.1. Pre-E-Beam Surface Treatment

After fabrication, a pre-E-beam surface treatment (adapted from Bruyas et al. [23]) was administered to all samples to make their surfaces more hydrophilic and rough, which facilitate better degradation solution penetration [29]. Samples were fully immersed in a 5 M NaOH (Ricca Chemical, Arlington, TX, USA) solution from diluting a 10 M stock with purified water (Milli-Q, MilliporeSigma, Burlington, MA, USA) at room temperature for 1 h (40% and 60% β-TCP) and 6.5 h (0% and 20% β-TCP). After NaOH submersion, all samples were rinsed twice with Milli-Q water and dried overnight. Under a sterile biological hood, the samples were then immersed in 70% ethanol for 20 min for sterilization, and then rinsed with PBS (pH 7.4, Gibco, Carlsbad, CA, USA) three times. After drying overnight, the samples were packaged in autoclaved self-sealing sterilization pouches under the sterile biological hood.

#### 2.1.2. E-Beam Specification

Half of all the samples were E-beam irradiated with a standard single dose of 25kGy (Steri-Tek, Fremont, CA, USA), in alignment with the ISO 11137-2:2006 norm. Steri-Tek uses two 10 MeV, 20 KW linear accelerators (Mevex, Stittsville, ON, Canada) to create a DualBeam™ processing method, which increases efficiency by administering uniform doses to products without having to rotate them. The Bruyas et al. study on E-beam and β-TCP/PCL scaffolds also used this E-beam specification [23]. This standard complies with the sterility assurance level (SAL) being less than 10^−6^. In other words, there can be at most one unsterile item for every one million objects, whether it be devices or scaffolds, in order to qualify as sterile [22].

### 2.2. The Subcutaneous Implantation of Samples into Rats

For this study, five Sprague Dawley rats (S.D Rat, Taconic Biosciences, Rensselaer, NY, USA) were grown in a pathogen-free environment for a period of 9 weeks. All experiments were conducted in accordance with animal testing ethics and were approved by Chonnam National University Institutional Animal Care and Use Committee (No. CNU IACUC-YB-2018-80). Specimens prepared for in vivo testing were classified as shown in Table 1, and a total of 40 specimens were prepared and divided into eight groups. The rats were anesthetized using 10 mg/kg of Xylazine (Rumpoon, Bayer, Leverkusen, Germany) and 20 mg/kg of Zoletil (Zolazepam + Tiletamine, Virbac, Carros, France) by intraperitoneal injection. To prevent bradycardia, 0.1 mg/kg of an anticholinergic drug (Atropine, JEIL Pharmaceutical, Seoul, Korea) was injected intramuscularly. Both the neck and hind limbs were shaved followed by iodine cure, ethanol (70% ethyl alcohol) disinfection, and incisions. Each filament from the experimental groups was implanted into the neck and hind limbs. Each rat was implanted with eight different groups of cylindrical filaments that were placed in subcutaneous sacs internally and sutured (Vicryl-4.0, Johnson & Johnson Medical, New Brunswick, NJ, USA), with sutures at appropriate intervals to prevent movement of the samples. The transplanted samples were not in contact with each other (Figure 1).

Post-operatively, all the animals received 5 mg/kg of antibiotics (Enrofloxacin, Bayer Leverkusen, Germany) and 5 mg/kg of analgesic anti-inflammatory drugs (Ketoprofen, EagleVet, Seoul, Korea). After implantation, rats were subjected to in vivo live micro-computer tomography (CT) (live-CT) and micro-CT (Figure 1).

### 2.3. Micro-CT and In Vivo Live-CT

Volumes of the samples before implantation and explants after animal euthanization were measured using micro-CT (SKYSCAN 1272, Bruker, Billerica, MA, USA) at 30 kV voltage, 150 μA current, and 10 μm pixel size. The scanned slices were reconstructed into DICOM files using the Cone Beam program (PerkinElmer, Waltham, MA, USA). After 24 weeks of implantation, the rats were sacrificed, and the implants were removed and subjected to micro-CT measurement under the same conditions as before implantation. The volume change of the specimens was calculated using Equation (1) and an image processing software (Mimics software, Materialize NV, Leuven, Belgium).
Micro-CT volume change (%) = ((*M*_△_ − *M*_0_)/*M*_0_) × 100,(1)
where *M*_0_ is the micro-CT data of implants before implantation, and *M*_△_ is the micro-CT data of explants at 24 weeks after implantation.

The volume changes of the filaments implanted in vivo were measured using a live-CT device (Quantum GX2, PerkinElmer, Inc., USA). Exposure conditions were maintained at 90 kV voltage, 88 µA current, and 90 µm voxel size for 4 min. The volume of the samples was measured after 1 day of implantation and again at 4, 12, and 20 weeks after implantation. The volume change of the specimens was calculated by Equation (2) using an image processing software.
Live-CT volume change (%) = ((*L*_△_ − *L*_0_) / *L*_0_) × 100,(2)
where *L*_0_ is the live-CT data at 1 day after implantation, and *L*_△_ is the data for each set period.

Equation (3) was used to compare the difference between micro-CT (*M_CT_*) volume before implantation and live-CT (*L_CT_*) volume after 1 day of implantation.
Difference between *L_CT_* and *M_CT_* (%) = ((*L_CT_* − *M_CT_*))/*M_CT_*) × 100.(3)

### 2.4. Histological Examination

Specimens were fixed using 4% paraformaldehyde solution and demineralized using 10% ethylenediaminetetraacetic acid (EDTA, Sigma-Aldrich, USA). The demineralized specimen was then dehydrated with increasing ethanol concentration (70% to 100%). Subsequently, it was cleaned with Xylene (Sigma-Aldrich, USA), embedded in paraffin, and then cut into 3 μm specimen slices using an Automated Rotary Microtome (Leica RM2255, Leica Microsystems, Wetzlar, Germany). Following the above process, the hydration step was executed. H&E staining (hematoxylin and eosin, Sigma-Aldrich, USA) was performed to evaluate histological characteristics including inflammatory response, collagen presence, and neovascularization.

### 2.5. Statistics

Quantitative data are presented as mean ± standard deviation with variance analysis according to the Mann–Whitney U test. The Kruskal–Wallis test was used to compare among experimental groups using the PASW Statistics 18.0 software (SPSS Inc., Chicago, IL, USA). p < 0.05 was considered statistically significant.

## 3. Results and Discussion

### 3.1. Evaluation of Filament Degradation in the Subcutaneous

Figure 2a shows the volume changes of the eight different groups of filaments before and after 24 weeks implantation by micro-CT analysis. The volume change of filaments was used as an indicator of degradation. At 24 weeks, the volume change of groups (0, e) and (0, no e) was 1.54% ± 0.28% and 0.78% ± 0.24%, respectively. Groups (20, e) and (20, no e) was 10.16% ± 3.95% and 6.58% ± 1.04%, respectively. Groups (40, e) and (40, no e) was 10.33 ± 4.19 and 5.68 ± 3.17, respectively. Groups (60, e) and (60, no e) was 10.47 ± 3.59 and 6.80 ± 1.54, respectively. There was a significant difference in volume change between the (0, e) and (0, no e) groups, but there was no significant difference among the other groups because of relatively high standard deviations.

However, the volume changes of E-beam groups were greater than the non-E-beam groups in the same TCP:PCL ratio pair. The average difference between the E-beam and non-E-beam groups in each ratio was 0.76% (0% TCP), 3.30% (20% TCP), 4.65% (40% TCP), and 3.67% (60% TCP). The results suggest that E-beam sterilization accelerated degradation, which is consistent with our in vitro degradation study [23] and other literature [19,20,23,26,27]. E-beam accelerates degradation because irradiation causes decreased crystallinity and shorter molecular chains of PCL due to chain scissions. The volume changes of pure PCL filaments are smaller than those of TCP/PCL composite filaments, suggesting that the addition of TCP also accelerated degradation. Compared to the slowly degradable hydrophobic PCL, the higher content of hydrophilic TCP ceramic particles increased water absorption from body fluids and accumulated more lipase enzymes onto the surface layer of the filaments [30,31], inducing hydrolysis and increasing PCL surface erosion [32]. It is worth noting that the increased hydrophilicity of the TCP/PCL surface likely facilitates cell adhesion, which also likely contributed to the increased degradation [32].

There were no significant differences in the degradation among the three different TCP content groups, in both E-beam and non-E-beam (Figure 2b,c). The results suggest that the chemistry of composites plays a bigger role in biodegradation than the sterilization method. In particular, E-beam mainly affected the properties of PCL, not TCP. The concentration of TCP particles embedded into the PCL matrix was not sufficient to allow particles to be interconnected. The lack of interconnection then allows water to penetrate into the center of the composite [19].

In this experiment, live-CT was used to observe the volume change of filaments over time without euthanizing the rats. Using live-CT also reduces the number of rats needed for the study and discrepancies due to the individual characteristics of each rat. However, the 0% TCP group was difficult to scan with live-CT due to its low contrast and fine fiber diameter. The ability to scan with live-CT is affected by the signal-to-noise ratio, the phase difference, and the tissue around the artifact [33,34,35,36].

Table 2 lists the volume differences of samples that were estimated by Equation (3) by subtracting the volume before implantation by micro-CT from the volume at one day after implantation by live-CT. The average volumes by live-CT were slightly higher than those by micro-CT in all samples. This difference could be due to the filament swelling inside the live body at one day after implantation. A study found that swelling of PCL occurs rapidly within the first 24 h, with a difference of about 10–20% [37]. In another report, the rate of swelling varies with the amount of hydrophilic material mixed with PCL within 24 h, and the resulting volume increase is around 5–10% [38]. Our results in Table 2 are consistent with the values in these reports.

Figure 3A shows the volume change over time based on live-CT analysis and Equation (2). All TCP-containing groups exhibited a rapid volume reduction during the first four weeks of implantation. From 4 to 20 weeks, all groups showed a slow and gentle downward slope in volume change, and the E-beam groups resorbed more at every time point. It has been reported that the adsorption of bioactive proteins to the surface of biomaterials from serum and body fluid upon implantation affects the rate of degradation [39], because this influences the effects of cellular interactions on body fluids and active protein substances attached to implanted TCP/PCL specimens. The hydrolysis mechanism of PCL occurred simultaneously, which results in rapid degradation [40].

However, one weakness of this study was that we did not use live-CT to examine the volumes of filaments at 24 weeks after implantation, while they were still implanted into the rats; rather, we used micro-CT to examine the volumes of the filaments after explantation. Our decision was misled by our data comparison at day one listed in Table 2, in which the volume by live-CT one day after implantation was greater than the volume by micro-CT before implantation. It turns out that the volumes from micro-CT analysis of the explants after 24 weeks of implantation are actually larger than the values from live-CT analysis at 20 weeks of implantation. The lack of live-CT data at 24 weeks after implantation lost us the opportunity to continuously examine changes between 20 and 24 weeks. As such, we cannot compare the difference between live-CT data before euthanization and micro-CT data after euthanization to further confirm the resolution of CT scanning.

### 3.2. Histological Examination

Figure 4 shows histology images of sample implants surrounded by tissue. Fibrous encapsulation—a thick and homogeneous colonization by fibroblast cells—was found in all samples.

Inflammatory macrophages (or monocytes) were found in the (0, e), (0, no e), (60, e), and (60, no e) groups. PCL degradation increases the acidity around the tissues, resulting in an inflammatory response [31]. Fibroblasts were at high-density and accompanied by collagen deposition. Blood vessel formation was also observed. A study of biodegradable polymers by Pêgo et al. shows that encapsulation of samples is mainly composed of macrophages, fibroblasts, and newly formed blood vessels due to immune reactions that are followed by inflammatory reactions, encapsulation characteristics commonly seen after implantation [41]. On the tissue surface of both the E-beam and non-E-beam 60% TCP groups, some multinucleated giant cells (MNGCs) were also observed (data not shown). When macrophages fail to remove foreign bodies due to the presence of slowly degradable PCL, they fuse to form MNGCs, and exhibit foreign body reactions during chronic inflammation [42]. The presence of MNGCs is closely related to the improvement of neovascularization, the degradation and uptake of the implanted biomaterial, and the encapsulation by the transplantation reaction [43,44]. MNGCs invade the implanted biomaterial and begin to destroy the original structure. Blood vessels and connective tissue grow into the biomaterial, leading to premature loss [45].

We understand that subcutaneous implantation may not reflect the degradation of orthotopic bone defect implantation. We used the filaments and subcutaneous implantation as a simple screening test before we test them in bone defects. In our other studies, we implanted 3D-printed TCP/PCL scaffolds of single formulation (20% TCP/80% PCL) into large bone defects in rabbit femoral heads [13,14] and rat femurs [46]. Both implantations showed that our scaffolds promoted bone ingrowth into the porous structure and retained structural integrity, suggesting excellent biocompatibility and osteointegration. The degradation rates of the macroporous 20% TCP/80% PCL scaffolds ranged from 10% to 25% at eight weeks after implantation in femoral head bone defects [13]. The slowest degradation rate in bone defects in rabbits was similar to those of subcutaneous implantation. The difference could be the result of higher surface areas in porous scaffolds versus rods and anatomical sites. We plan to further test E-beam sterilized 3D-printed scaffolds within bone defects in the future.

## 4. Conclusions

In this study, we extruded β-TCP/PCL filaments of different ratios and sterilized the samples with clinically available E-beam irradiation, and implanted them in subcutaneous sites for 24 weeks. The degradation rates were characterized by micro-CT and live-CT, and biocompatibility of samples was examined by histology analyses. We observed that incorporation of TCP into PCL significantly increased degradation of the composite, but increasing TCP content in the composite did not accelerate degradation. Faster degradation occurred in the first four weeks and gradually slowed down afterward. E-beam sterilization also accelerated degradation, which is due to decreased crystallinity and shorter molecular chains of PCL after the E-beam irradiation. For the TCP/PCL filaments, the chemistry of samples plays a bigger role than the sterilization method in biodegradation. E-beam sterilization did not affect biocompatibility of the implants in the subcutaneous implantation. Our work suggests that creating TCP/PCL composites is a promising method for achieving the degradation properties required to fulfill clinical demands for the use of E-beam sterilization in osteosynthetic applications.

## Figures and Tables

**Figure 1 micromachines-11-00273-f001:**
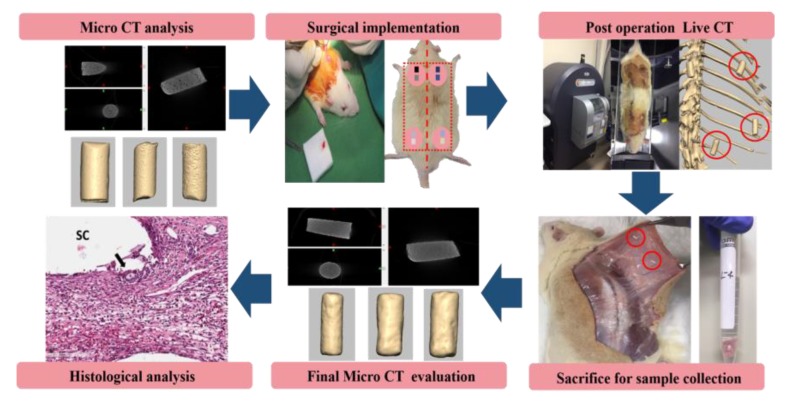
Schematic diagram of the filament implantation and experimental process.

**Figure 2 micromachines-11-00273-f002:**
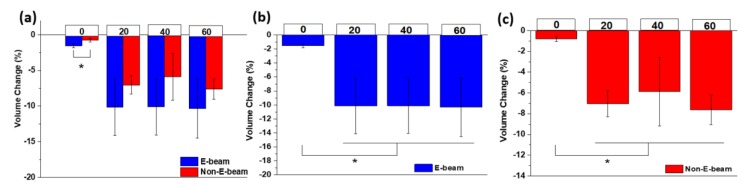
Volume change of filaments by micro-computer tomography (CT) and Equation (1). (**a**) Between E-beam and non-E-beam at 24 weeks (* p < 0.05); (**b**,**c**) of 4 different tricalcium phosphate/polycaprolactone (TCP/PCL) ratio groups at 24 weeks (* p < 0.05).

**Figure 3 micromachines-11-00273-f003:**
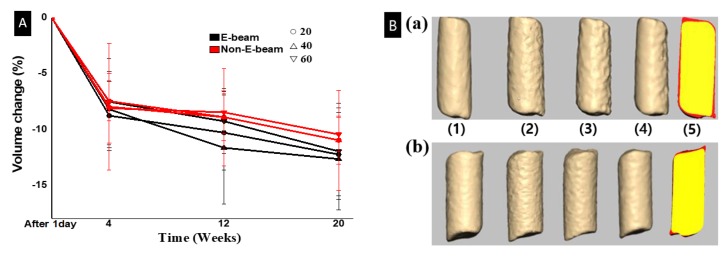
(**A**) Volume change at 1 day, 4, 12, and 20 weeks after implantation by live-CT and Equation (2). (**B**) Three-dimensional model of a sample 40% TCP filament, (**a**) E-beam and (**b**) non-E-beam by live-CT. (1–4) are filaments at 1 day, 4 weeks, 12 weeks, and 20 weeks, respectively. (5) is a superimposition between 1 day (red) and 20 weeks (yellow).

**Figure 4 micromachines-11-00273-f004:**
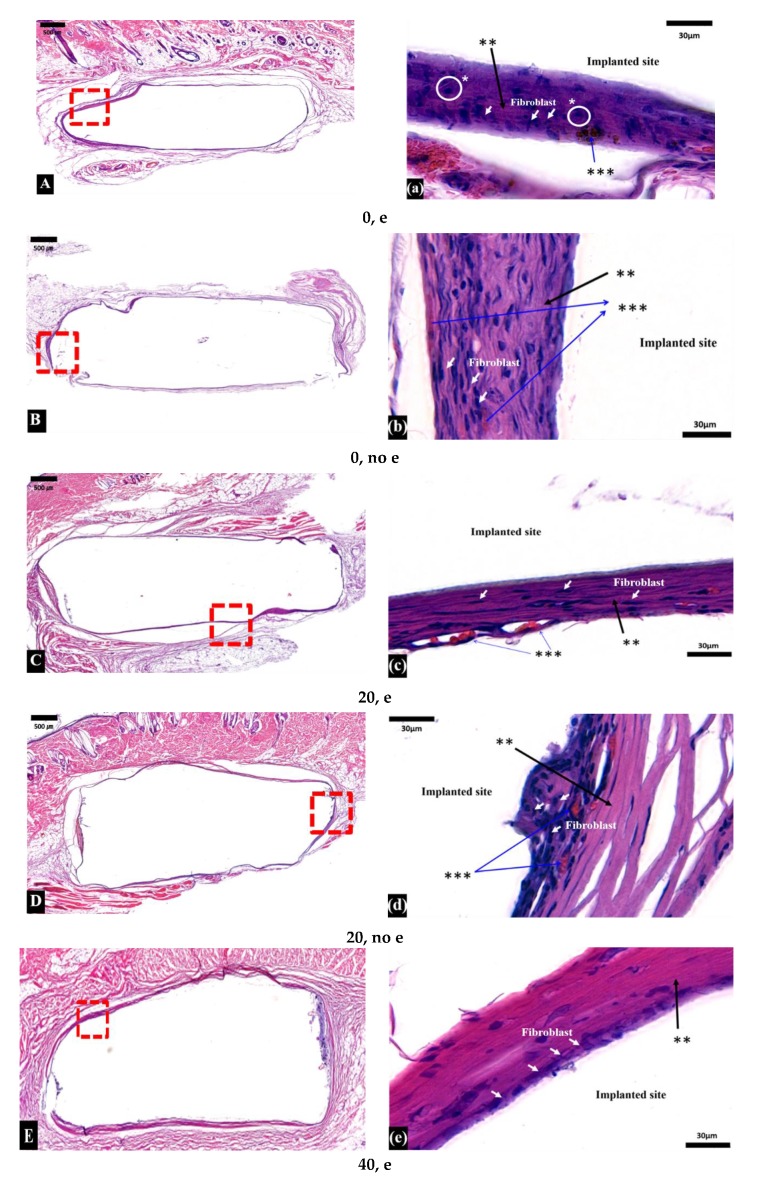
Micrographs of subcutaneous tissue responses to filaments after 24 weeks of implantation. White arrows indicate fibroblasts; (* with a white circle) indicates monocytes and macrophages; (**) indicates protein and collagen; and (***) indicates blood vessels. The micrographic images of each stained samples (10 × and 100 × magnification) are labelled with group code as follow; (**A**) and (**a**) are (0,e), (**B**) and (**b**)are (0,no e), (**C**) and(**c**) are (0,e), (**D**) and(**d**) are(0,no e), (**E**)and (**e**)are(0,e), (**F**)and (**f**)are (0,no e), (**G**) and (**g**)are (0,e),and (**H**) and (**h**)are(0,no e).

**Table 1 micromachines-11-00273-t001:** Classification codes for each group.

Filament Group		Code	Quantity
E-beam	100% PCL	0, e	5
	20% TCP/80% PCL	20, e	5
	40% TCP/60% PCL	40, e	5
	60% TCP/40% PCL	60, e	5
Non-E-beam	100% PCL	0, no e	5
	20% TCP/80% PCL	20, no e	5
	40% TCP/60% PCL	40, no e	5
	60% TCP/40% PCL	60, no e	5

**Table 2 micromachines-11-00273-t002:** Volume difference between micro-CT and live-CT.

Time/Group	20, e	40, e	60, e	20, no e	40, no e	60, no e
Before surgery (micro-CT) (%)	28.05	21.30	27.73	26.13	22.00	26.24
1 day after implantation (live-CT) (%)	28.83	22.64	29.23	26.93	23.74	27.25
Volume change (%)	+2.78	+6.29	+5.4	+3.09	+7.88	+3.84

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
