# Peer review of "The Influence of Electron Beam Sterilization on In Vivo Degradation of β-TCP/PCL of Different Composite Ratios for Bone Tissue Engineering"

_micromachines, 2020, doi:10.3390/mi11030273_

Round 1

Reviewer 1 Report

The authors report the effect of electron beam (E-beam) sterilization on the degradation of β-tricalcium phosphate / polycaprolactone (β-TCP/PCL) composite filaments with various ratios (0:100, 20:80, 40:60, and 60:40 by mass) in a rat subcutaneous model for 24 weeks and the results suggest that E-beam groups showed greater volume reduction at every time point. The study is interesting and there are a few comments that need to be addressed:

  1. What is the real novelty in this study? There has been many studies as stated in the introduction [19, 20, 23, 26, 27] which suggests that E-beam has an effect on degradation (accelerates it). What is the novelty in this study, except that it is done in vivo?
  2. Why E-beam alone is chosen as the sterilization method? We have studied PCL extensively and have used UV-sterilization (papers below). Could you compare the below papers and also other papers with different sterilization methods (just dipping in ethanol, or other solvents, etc.) in the introduction to make a case why E-beam is considered? (a) "3D-Printed PCL/PPy Conductive Scaffolds as Three-Dimensional Porous Nerve Guide Conduits (NGCs) for Peripheral Nerve Injury Repair." Frontiers in bioengineering and biotechnology 7 (2019): 266. (b) "Electrohydrodynamic jet 3D printed nerve guide conduits (NGCs) for peripheral nerve injury repair." Polymers 10.7 (2018): 753.
  3. Could you get more quantitative data from the histological staining (H&E) images to make the results more authentic and informative?
  4. The line graphs in Figures 2 & 3 are low-resolution, image quality has to be improved.

Author Response

Following your decision letter on the above-said manuscript submitted for publication, we are sending our response letter addressing in detail the questions and issues raised by the reviewer and the changes performed on the manuscript, as per your suggestions. Firstly, we are thankful to the reviewers and editors for providing a favorable response for the publication of our manuscript in this esteemed journal. We wish to state that we have carefully gone through every comment and issue raised and have taken sincere efforts to clarify and incorporate the suggestions of the reviewers.
The answers to the reviewer questions were combined with the revised manuscript and submitted in a PDF file.

Reviewer 2 Report

The manuscript submitted by Kang et al. reports on the characterization of electron beam sterilization on in vivo degradation of β-TCP/PCL blends for bone tissue engineering applications. The results are novel enough and suitable for the readership of Micromachines. Nonetheless, several points must be addressed before being suitable for publication: 

  1. In general this reviewer suggests to have the manuscript proofread by an English native speaker to reformulate some sentences whose grammar is not impeccable.
  2. Line 54/55: what do the authors mean with “ease of use”? Please be more specific.
  3. Line 77: can the author compare E-beam sterilization with other standard procedure for scaffold sterilization such as UV irradiation or ethanol washes?
  4. Line 85: Which were the values of Young’s modulus of the scaffolds not treated with E-beam? How were these measurements performed?
  5. Line 96/97: the authors choose to use extruded filaments for their in-vivo studies but, especially for bone tissue engineering, the role of porosity in the scaffolds is crucial and can also influence degradation. Please comment.
  6. Line 130-134: the description of the E-beam setup is rather minimalistic. Please provide additional details (e.g. system name, beam size etc…).
  7. Line 145-147: It is not clear in which regions of the rat models the betatcp/PCL cylindirical filaments were implanted. Which is the rationale behind the choice of the specific locations?
  8. Line 203-205: Figures 2A,B,C,D seem redundant. This reviewer suggests to keep only Figure 2B.

Author Response

(The authors gave the same response as above.)

Round 2

Reviewer 1 Report

All comments are addressed. With reference to comment #2 of Reviewer 1, Ref #22 is a more general reference of sterilization methods, the recommended references are specific to PCL, the authors may choose not to cite the recommended references for whatever reason but at least references that are relevant to PCL scaffold sterilization, as E-beam is not commonly used for sterilizing PCL. 

Reviewer 2 Report

The authors have addressed all my remarks.